# Allosteric modulation of the adenosine A$_{2A}$ receptor by cholesterol

**Shuya Kate Huang**[1,2], **Omar Almurad**[1,2], **Reizel J Pejana**[1,2], **Zachary A Morrison**[1], **Aditya Pandey**[1,2], **Louis-Philippe Picard**[1,2], **Mark Nitz**[1], **Adnan Sljoka**[3,4], **R Scott Prosser**[1,2,5]*

[1]Department of Chemistry, University of Toronto, Toronto, Canada; [2]Department of Chemical and Physical Sciences, University of Toronto Mississauga, Mississauga, Canada; [3]RIKEN Center for Advanced Intelligence Project, Tokyo, Japan; [4]York University, Department of Chemistry, Toronto, Canada; [5]Department of Biochemistry, University of Toronto, Toronto, Canada

**Abstract** Cholesterol is a major component of the cell membrane and commonly regulates membrane protein function. Here, we investigate how cholesterol modulates the conformational equilibria and signaling of the adenosine A$_{2A}$ receptor (A$_{2A}$R) in reconstituted phospholipid nanodiscs. This model system conveniently excludes possible effects arising from cholesterol-induced phase separation or receptor oligomerization and focuses on the question of allostery. GTP hydrolysis assays show that cholesterol weakly enhances the basal signaling of A$_{2A}$R while decreasing the agonist EC$_{50}$. Fluorine nuclear magnetic resonance ($^{19}$F NMR) spectroscopy shows that this enhancement arises from an increase in the receptor's active state population and a G-protein-bound precoupled state. $^{19}$F NMR of fluorinated cholesterol analogs reveals transient interactions with A$_{2A}$R, indicating a lack of high-affinity binding or direct allosteric modulation. The combined results suggest that the observed allosteric effects are largely indirect and originate from cholesterol-mediated changes in membrane properties, as shown by membrane fluidity measurements and high-pressure NMR.

*For correspondence: scott.prosser@utoronto.ca

**Competing interest:** The authors declare that no competing interests exist.

## Editor's evaluation

Cholesterol has long been known to have significant effects on G protein-coupled receptor (GPCR) ligand binding properties and stability, and cholesterol/GPCR interactions have frequently been observed in high-resolution structures. However, relatively limited biophysical work has been done to investigate the mechanistic basis for cholesterol's effects. This manuscript describes the use of a sensitive 19F NMR probe to monitor conformational equilibria in a prototypical GPCR, the A2a adenosine receptor. These experiments, together with data from other NMR experiments, computational analysis, and G protein assays, show that the subtle effects of cholesterol derive in large part from modulation of membrane biophysical properties, in contrast to conventional allosteric modulators that exert their effects through direct long-lived receptor binding.

## Introduction

In mammalian cell membranes, cholesterol accounts for ~5–45% of the total lipid content across different cell types and subcellular components (*Casares et al., 2019*; *Ingólfsson et al., 2017*). It is a critical metabolic precursor to steroid hormones, bile salts, and vitamin D, while numerous cardiovascular and nervous system disorders are attributed to abnormalities in cholesterol metabolism (*Arsenault et al., 2009*; *Martín et al., 2014*). The rigid planar structure of cholesterol promotes ordering of

bilayer lipids, thus modulating membrane fluidity and thickness. Cholesterol also drives the formation of raft-like microdomains and commonly interacts with membrane proteins as a ligand or allosteric modulator (*Hulce et al., 2013*).

Here, we investigate how cholesterol influences the conformational equilibria and signaling of a well-studied integral membrane protein, the adenosine $A_{2A}$ receptor ($A_{2A}R$), in reconstituted phospholipid/cholesterol nanodiscs. Specifically, we seek to understand if the effects on $A_{2A}R$ function are a consequence of direct allosteric interplay between cholesterol and the receptor, or if the observed effects result primarily from cholesterol-driven changes in viscoelastic properties and thickness of the lipid bilayer.

$A_{2A}R$ is a member of the rhodopsin family of G-protein-coupled receptors (GPCRs). The GPCR superfamily of 7-transmembrane receptors includes well over 800 species and are targeted by over one-third of currently approved pharmaceuticals (*Hauser et al., 2017*). $A_{2A}R$ activates the stimulatory heterotrimeric G protein ($G_s\alpha\beta\gamma$) and is a target for the treatment of inflammation, cancer, diabetes, and Parkinson's disease (*Effendi et al., 2020*; *Guerrero, 2018*; *de Lera Ruiz et al., 2014*; *Yu et al., 2020*; *Zheng et al., 2019*). Several GPCRs have been shown to interact with cholesterol, including the serotonin 5-$HT_{1A}$ receptor, the $\beta_2$-adrenergic receptor, the oxytocin receptor, the smoothened receptor, the CCR5 and CXCR4 chemokine receptors, the $CB_1$ cannabinoid receptor, and $A_{2A}R$ (*Gimpl, 2016*; *Jafurulla et al., 2019*; *Kiriakidi et al., 2019*). Presently, 38 out of 57 published structures of $A_{2A}R$ contain co-crystallized cholesterol (*Figure 1*). In detergent preparations of $A_{2A}R$, the soluble cholesterol analog, cholesteryl hemisuccinate (CHS), is important for receptor stability and ligand binding (*O'Malley et al., 2011a*; *O'Malley et al., 2007*). Apart from those found in crystal structures, cholesterol interaction sites within $A_{2A}R$ have also been proposed in computational studies. These include the widely conserved cholesterol consensus motif (CCM) in GPCRs, various hydrophobic patches around $A_{2A}R$, and regions of the receptor interior (*Genheden et al., 2017*; *Guixà-González et al., 2017*; *Lee et al., 2013*; *Lee and Lyman, 2012*; *Lovera et al., 2019*; *McGraw et al., 2019*; *Rouviere et al., 2017*; *Sejdiu and Tieleman, 2020*; *Song et al., 2019*). The CRAC (cholesterol recognition/interaction amino acid consensus) motif, another sequence commonly found in membrane proteins that bind cholesterol, is also present in $A_{2A}R$ (*Figure 1B*; *Li and Papadopoulos, 1998*). Additionally, cell-based assays have shown that $A_{2A}R$-dependent cyclic adenosine monophosphate (cAMP) production is positively correlated with membrane cholesterol (*Charalambous et al., 2008*; *McGraw et al., 2019*).

Despite the prevalence of cholesterol or its analogues in many crystal structures, there is little consensus on the role that membrane cholesterol plays in $A_{2A}R$ function. While some studies found that ligand binding was unaffected by cholesterol depletion (*Charalambous et al., 2008*; *McGraw et al., 2019*), others have observed opposite effects (*Guixà-González et al., 2017*; *O'Malley et al., 2011a*; *O'Malley et al., 2011b*). One study in particular suggested that cholesterol may laterally diffuse in the membrane and enter the receptor interior at the orthosteric site (*Guixà-González et al., 2017*). Additionally, whereas $A_{2A}R$ is found in both non-raft and raft-like membranes, its colocalization and modulatory effects on other cellular binding partners, including tyrosine receptor kinase B, $Ca^{2+}$-activated $K^+$ (IK1) channel, and the stimulatory G protein, have been reported to depend on cholesterol-rich microdomains (*Charalambous et al., 2008*; *Lam et al., 2009*; *Mojsilovic-Petrovic et al., 2006*). One possible source of discrepancy between studies is the use of different cell lines. For instance, Guixà-González et al. observed an increased binding by $A_{2A}R$ inverse agonist [3H]ZM241385 upon cholesterol depletion in C6 glioma cells. This effect was absent in a study by McGraw et al., who employed HEK293 cells. Cholesterol extraction or enrichment from cells exhibiting different membrane compositions and signaling patterns may trigger variable cellular response and complicates the comparison of results from different cell lines. The in vitro studies, on the other hand, relied on measuring ligand affinity in detergent micelles while titrating water-soluble cholesterol analogs. Although the composition of detergent preparations can be carefully controlled, the micellar environment is quite different from a lipid bilayer from the perspective of both receptor and cholesterol.

To mitigate the many complexities encountered in live cells or the inherent biases associated with detergent micelles, we employed reconstituted discoidal high density lipoprotein particles (rHDLs, also known as nanodiscs) to investigate the role of cholesterol in $A_{2A}R$ conformation and signaling. In this case, both the size and composition of these phospholipid bilayer model systems can be controlled. Through fluorine nuclear magnetic resonance spectroscopy ($^{19}F$ NMR) and in vitro assays,

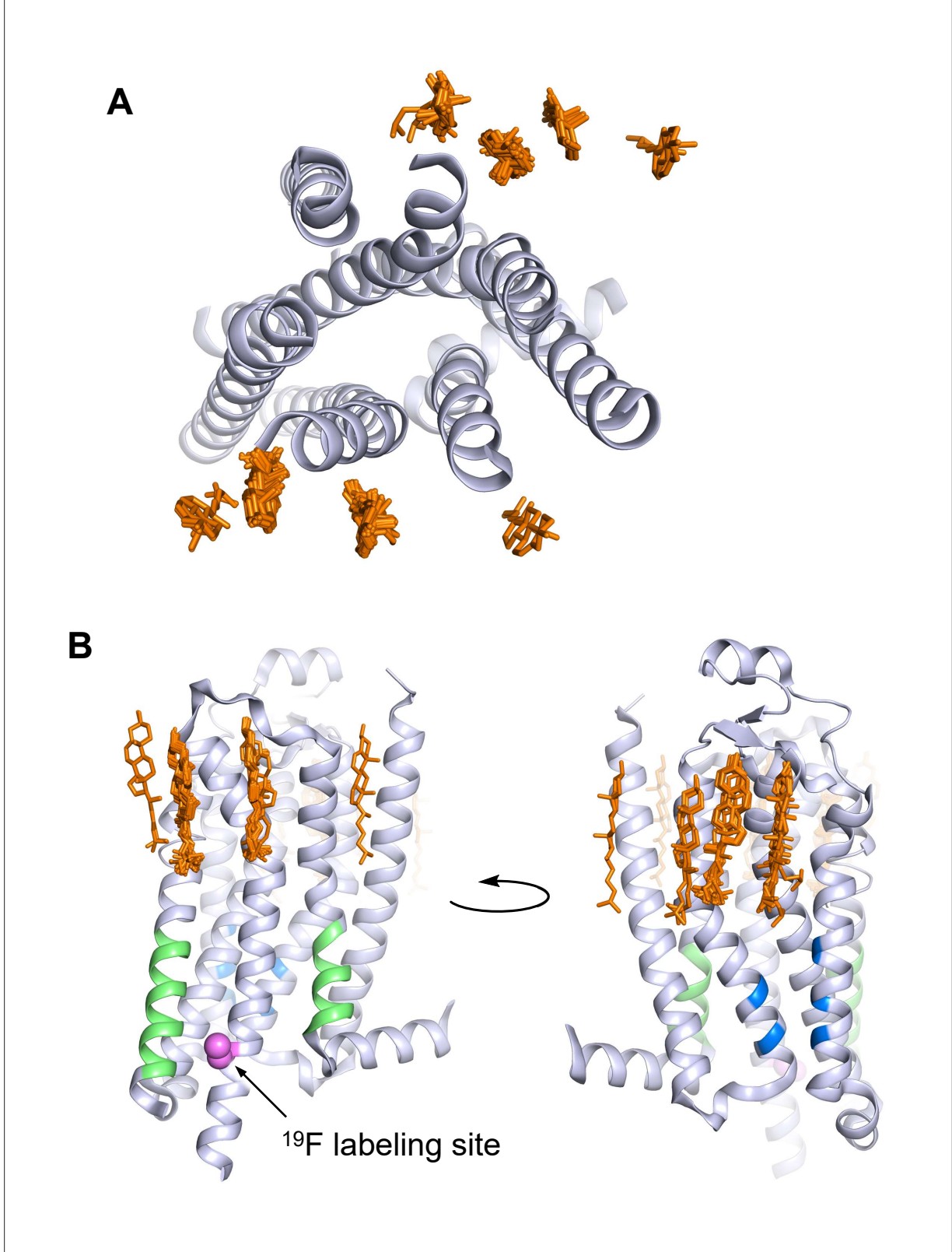

**Figure 1.** A$_{2A}$R crystal structures reveal many cholesterol interaction sites. (**A**) Overlay of 38 currently published A$_{2A}$R crystal structures containing co-crystallized cholesterol (extracellular view, with cholesterols shown as orange sticks). For simplicity, extracellular loops and fusion proteins are removed and only one receptor structure is shown (PDB: 4EIY). (**B**) Side views of (**A**) highlighting the CCM (blue), the CRAC motifs (green), and V229C $^{19}$F labeling site (violet).

we find that cholesterol is a weak positive allosteric modulator of $A_{2A}R$. This can be attributed to a subtle rise in population of the receptor's active state conformers and a stronger coupling to the G protein. Interactions between $A_{2A}R$ and fluorinated cholesterol analogs appear to be short-lived and non-specific, indicating a lack of high-affinity binding sites or direct allosteric modulation. Rather, the observed allostery is likely a result of indirect membrane effects through cholesterol-mediated changes in bilayer fluidity and thickness, which can be recapitulated (without the use of cholesterol) by the application of hydrostatic pressure.

## Results

### Cholesterol is a weak positive allosteric modulator of $A_{2A}R$

We sought to explore receptor-cholesterol allostery in a native lipid bilayer environment, free from the complexities associated with other cellular response to membrane alteration in live cells. To this end, we reconstituted $A_{2A}R$ (residues 2–317 with valine 229 mutated to cysteine for $^{19}F$-labeling) in nanodiscs containing a 3:2 ratio of 1-palmitoyl-2-oleoyl-*sn*-glycero-3-phosphocholine (POPC) and 1-palmitoyl-2-oleoyl-*sn*-glycero-3-phospho-(1'-*rac*-glycerol) (POPG), supplemented with different amounts of cholesterol. In our hands, cosolubilization of cholesterol with phospholipids prior to reconstitution (*Midtgaard et al., 2015*) resulted in polydisperse particles and low cholesterol incorporation. We therefore adapted a procedure commonly used in cells and liposomes, to deliver cholesterol via methyl-β-cyclodextrin (MβCD) to preformed nanodiscs (*Zidovetzki and Levitan, 2007*). This allowed us to incorporate up to ~15 mol% cholesterol into $A_{2A}R$-embedded nanodiscs without affecting their size distribution (*Figure 2—figure supplement 1*).

To examine the effects of cholesterol on receptor-mediated G-protein activation, we measured the GTPase activity of purified G proteins ($G_s\alpha_{short}\beta_1\gamma_2$, henceforth referred to as $G\alpha\beta\gamma$) in the presence of $A_{2A}R$-nanodiscs containing 0%, 3%, 8%, 11%, and 13% cholesterol. As shown in *Figure 2A*, similar agonist dose-response profiles were obtained across different cholesterol concentrations. GTP hydrolysis (cumulative over 90 min) was higher in the presence of $A_{2A}R$ relative to G protein alone and was amplified by the agonist 5'-*N*-ethylcarboxamidoadenosine (NECA) in a dose-dependent manner. Upon careful inspection, a small yet notable decrease in agonist $EC_{50}$ values can be observed as a function of cholesterol. There is also a slight enhancement in receptor basal activity at high cholesterol concentrations (*Figure 2B–C*). Thus, functionally cholesterol behaves as a positive allosteric modulator (PAM) of $A_{2A}R$, although there is very weak cooperativity between cholesterol and agonist.

The weak cholesterol dependence above implies that either cholesterol does not form tight interactions with $A_{2A}R$, or that the interactions it establishes with the receptor do not grossly overlap with the predominant allosteric pathways established by the agonist. The observed enhancement may also be a consequence of an indirect effect resulting from changes to membrane physical properties. Although the amounts of cholesterol used in this study were lower than that of a typical plasma membrane, they greatly exceed the concentrations needed to saturate potential high-affinity binding sites. Therefore, a simple allosteric mechanism involving specific binding by cholesterol is unlikely.

Using $^{19}F$ NMR, it is possible to directly assess the effects of cholesterol on the distribution of receptor functional states. Based on the agonist dose-response curves, we expected a stabilization of activation intermediates or active states, at least in the presence of G protein. $^{19}F$ NMR spectra of $A_{2A}R$ were recorded as a function of ligand, G protein, and cholesterol (*Figure 3A* and *Figure 3—figure supplement 1*). In this case, the receptor was labeled at the intracellular side of transmembrane helix 6 (TM6), in a region known to undergo large conformational changes upon activation (*Figure 3B*). The resulting resonances have been assigned in our previous works and are shown as cartoons in *Figure 3C* (*Huang et al., 2021*; *Ye et al., 2016*). Briefly, $S_1$ and $S_2$ represent two inactive state conformers differentiated by a conserved salt bridge ('ionic lock') between TM3 and TM6. The $A_3$ state is an activation intermediate stabilized by $G\alpha\beta\gamma$ binding in the absence of ligands and is hence associated with the 'precoupled' state. $A_1$ and $A_2$ represent distinct active state conformers that facilitate nucleotide exchange in the receptor G-protein complex. It was found that while $A_1$ is preferentially stabilized by full agonist, $A_2$ is more pronounced in the presence of partial agonist.

Inspection of the overlaid spectra in *Figure 3A* reveals nearly identical distributions of conformational states between $A_{2A}R$ in the presence of 0% and 4% membrane cholesterol. At 13% cholesterol, subtle changes can be observed in the inverse agonist-saturated, apo, full agonist-saturated, and full

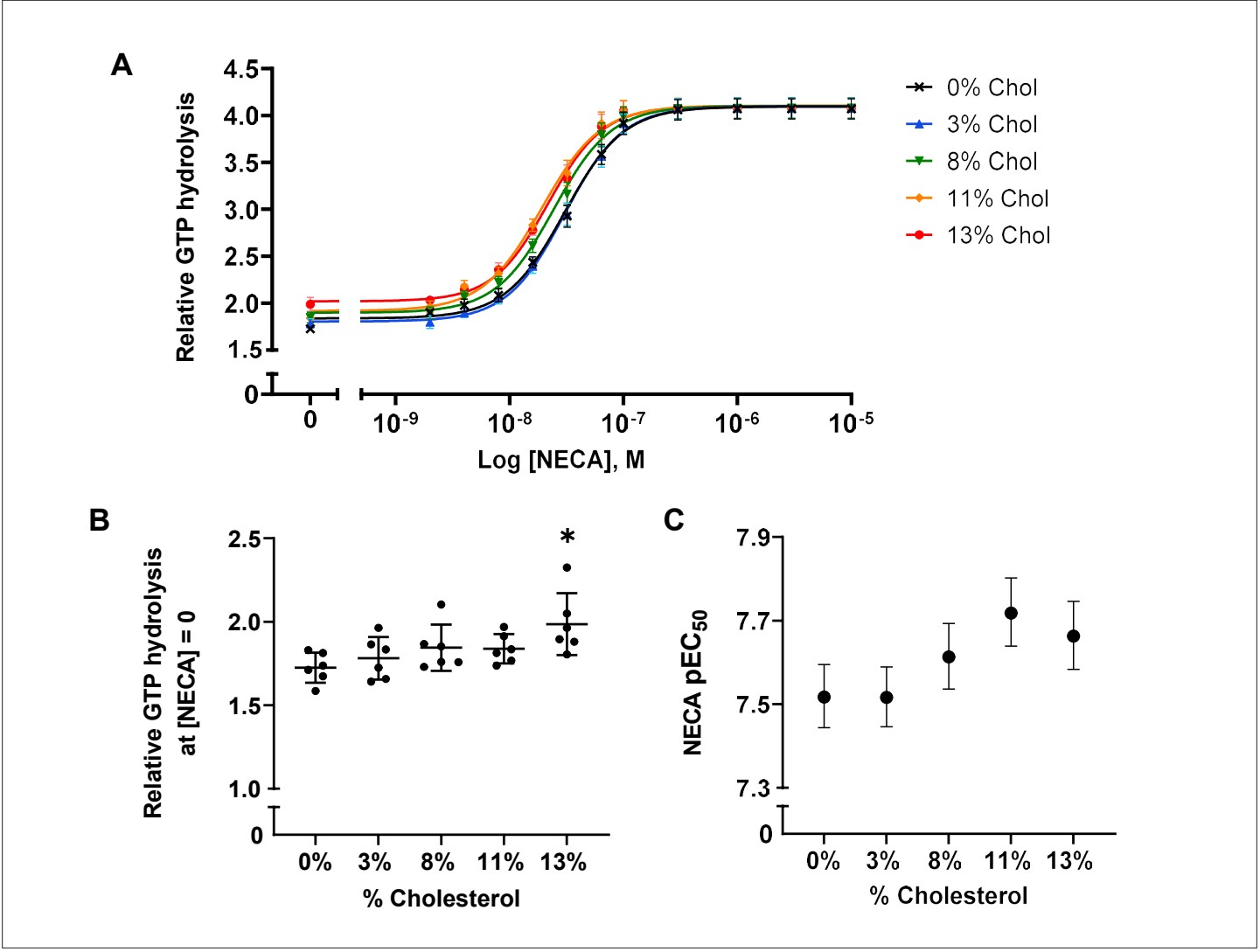

**Figure 2.** $A_{2A}R$ agonist potency and basal activity are weakly enhanced by cholesterol. (**A**) Agonist (NECA) dose-response curves for $A_{2A}R$-nanodiscs containing varying concentrations of cholesterol. The vertical axis represents GTP hydrolysis by purified $G\alpha\beta\gamma$ (cumulative over 90 min) in the presence of $A_{2A}R$ and agonist, relative to GTP hydrolysis by $G\alpha\beta\gamma$ alone in the absence of $A_{2A}R$. Each data point represents the mean ± SEM (n = 6, technical triplicates). (**B**) Relative GTP hydrolysis in the presence of apo $A_{2A}R$ (no agonist) in nanodiscs containing varying concentrations of cholesterol. Data represents mean ± SD (n = 6, averages from each technical triplicate presented as individual points) and the asterisk represent statistical significance relative to the 0% cholesterol condition. Statistical significance was determined by one-way ANOVA followed by the Tukey's multiple comparison test. * p ≤ 0.05. (**C**) $pEC_{50}$ values of the NECA dose-response curves in (**A**). Error bars represent 95% (asymmetrical profile likelihood) confidence intervals.

The online version of this article includes the following figure supplement(s) for figure 2:

**Figure supplement 1.** The size distribution of nanodiscs are minimally affected by cholesterol incorporation.

agonist+ $G\alpha\beta\gamma$ spectra. In particular, we observed a small population shift toward the active states, $A_1$ and $A_2$. The results imply that cholesterol is a positive allosteric modulator of $A_{2A}R$ and acts in part through stabilization of the active state ensemble. Interestingly, 13% cholesterol resulted in line-broadening and a 0.09 ppm downfield shift of the $A_1$ state in the full agonist-saturated spectrum, changes which are not observed in the full agonist+ $G\alpha\beta\gamma$ condition. The changes seen with the $A_1$ resonance in the presence of 13% cholesterol and NECA alone are likely a consequence of exchange between $A_1$ and the inactive ionic lock broken state ($S_2$) and/or a perturbation of the average orientation of TM6. Clearly, cholesterol exerts differential effects on distinct conformers in the ensemble. The outcome of the NMR experiments is consistent with the GTPase activity assay. Nearly complete overlap between the 0% and the 4% spectral series suggest that the principal allosteric mechanism is unlikely related to high-affinity binding. While the subtle changes observed at 13% cholesterol are

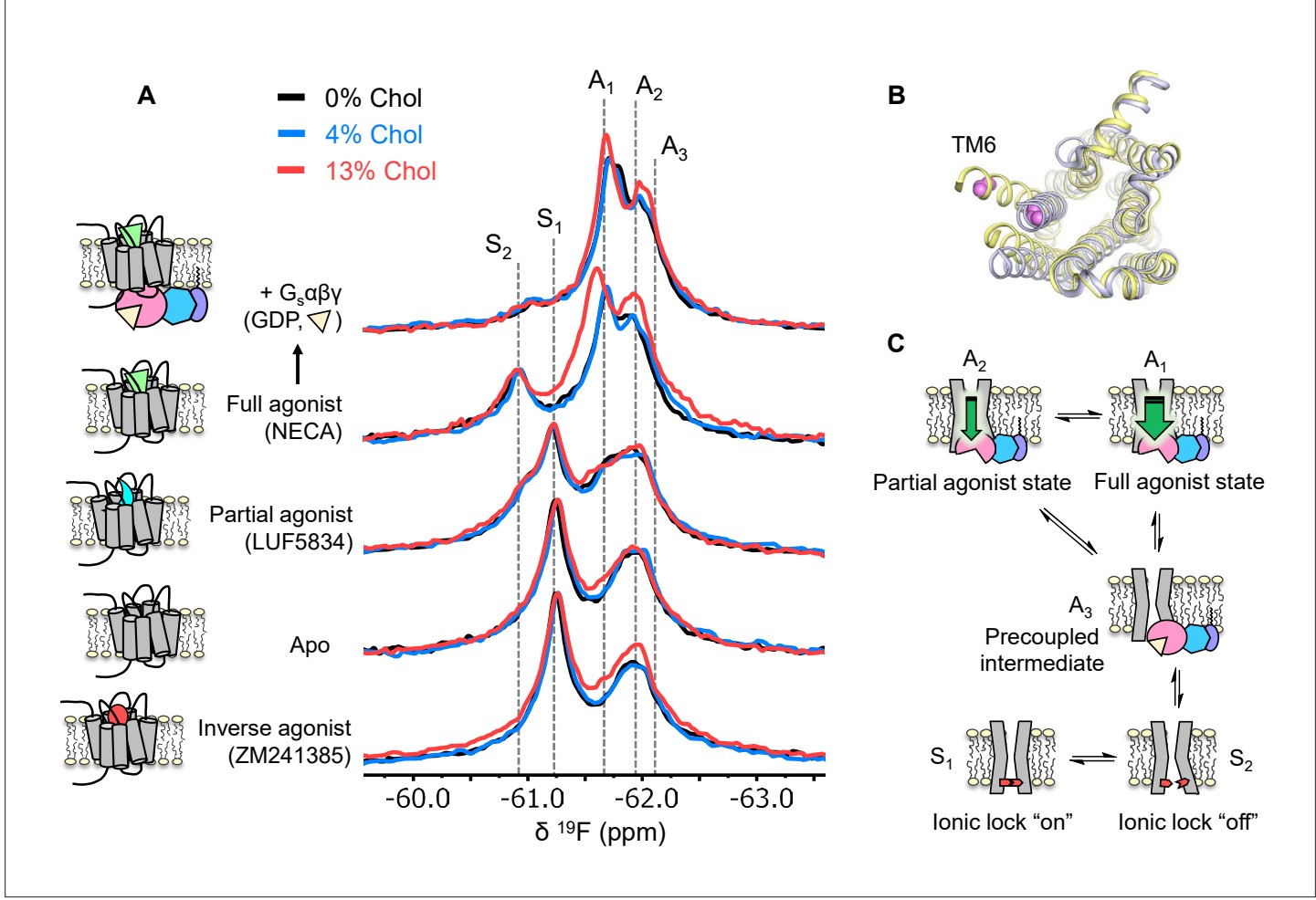

**Figure 3.** Cholesterol induces a small population increase in the active state conformers of $A_{2A}R$. (**A**) $^{19}F$ NMR spectra of $A_{2A}R$ in nanodiscs containing 0, 4, and 13% cholesterol, as a function of ligand and G protein. Two inactive states ($S_{1-2}$) and three active states ($A_{1-3}$), previously identified, are indicated by gray dashed lines. For each ligand condition, spectra from the three cholesterol concentrations are normalized via their inactive state intensity. (**B**) Intracellular view of an inactive (gray, PDB: 4EIY) and an active (yellow, PDB: 5G53) crystal structure of $A_{2A}R$ highlighting the movement of TM6 upon activation. The $^{19}F$-labeling site is shown in violet. (**C**) Cartoon representations of the key functional states of $A_{2A}R$ indicated in (**A**). At the bottom are two inactive states ($S_1$ and $S_2$) where a conserved salt bridge is either intact or broken. $A_3$ is an intermediate state that facilitates G-protein recognition and precoupling. $A_1$ and $A_2$ are active states that drive nucleotide exchange. While $A_1$ is more efficacious and preferentially stabilized by the full agonist, $A_2$ is less efficacious and reinforced by a partial agonist.

The online version of this article includes the following figure supplement(s) for figure 3:

**Figure supplement 1.** $A_{2A}R$ exhibits similar activation signatures in the absence and presence of cholesterol.

evidence for positive allosteric modulation, the effects are much smaller than those of any orthosteric ligands or other known allosteric modulators of $A_{2A}R$ (**Gao et al., 2020**; **Ye et al., 2018**).

To understand if the weakly activating role of cholesterol arises because of enhanced efficiency in nucleotide exchange or pre-association with G protein (precoupling), we carried out $^{19}F$ NMR experiments on apo-$A_{2A}R$ in the presence of $G\alpha\beta\gamma$ without agonist. As mentioned above, this condition produces the precoupled receptor-G protein complex and greatly stabilizes the $A_3$ state (**Huang et al., 2021**). This is recapitulated in **Figure 4** for all three cholesterol concentrations, where a shift in the equilibrium populations toward the active conformers, particularly $A_3$ and $A_2$, is observed upon the addition of $G\alpha\beta\gamma$. Importantly, an increase in cholesterol further enhanced the $A_3$ state in addition to a decrease in the peak width. The magnitudes of these changes are small, consistent with results shown in **Figures 2–3**. The results suggest that membrane cholesterol may help to stabilize the precoupled complex of $A_{2A}R$ and G protein, and possibly modulates the amplitudes of motion about the precoupled state. This in turn may favor further conformational exchange to $A_1$ or $A_2$. Taken together, $^{19}F$ NMR

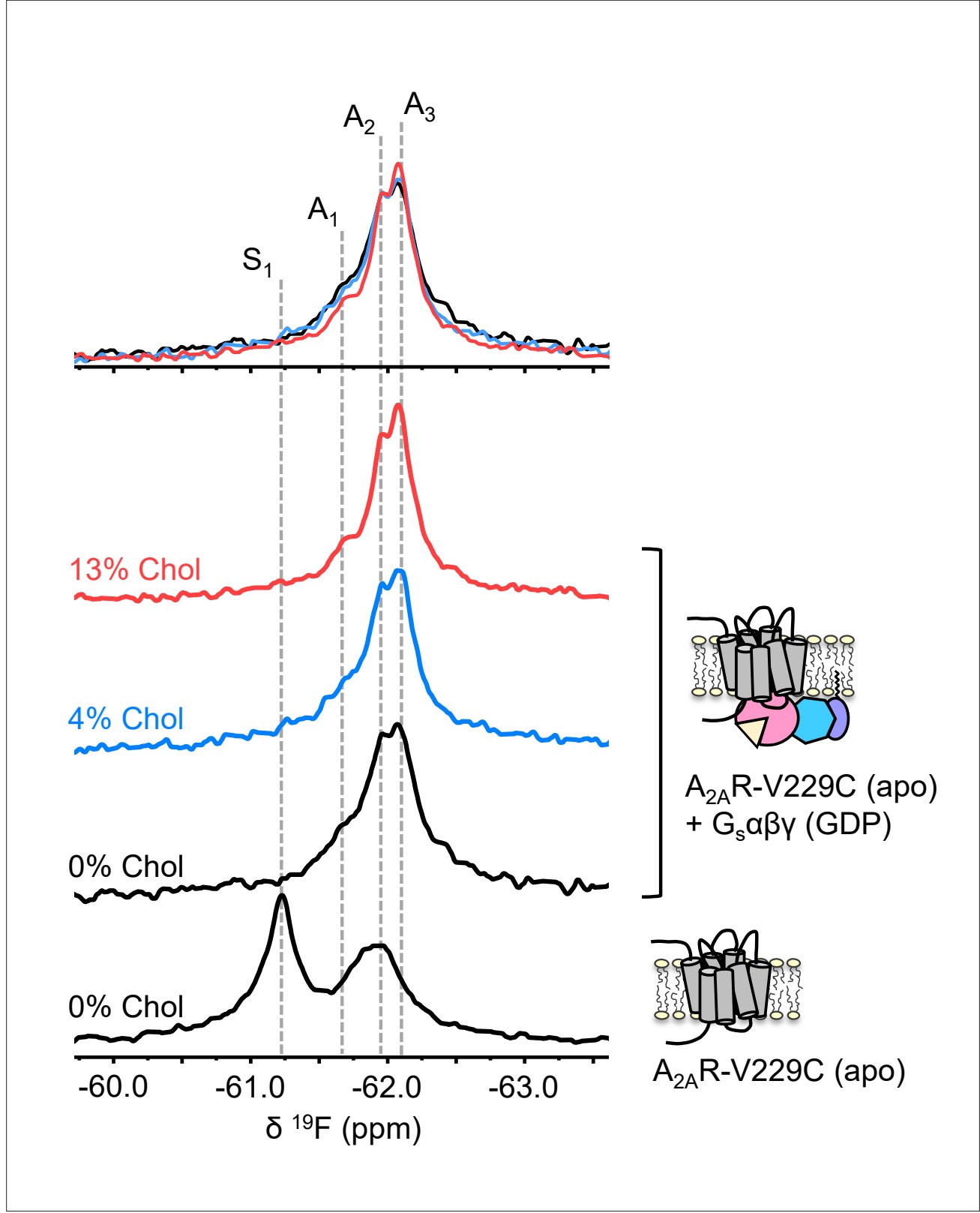

**Figure 4.** The precoupled complex of $A_{2A}R$-$G_s\alpha\beta\gamma$ is stabilized by cholesterol. $^{19}F$ NMR spectra of apo $A_{2A}R$ in the presence of 0, 4, and 13% cholesterol, and as a function of $G\alpha\beta\gamma$. The key functional states are indicated by gray dashed lines and the three spectra in the presence of G protein are normalized via the $A_2$ state.

showed that mechanistically, the PAM effect of cholesterol in $A_{2A}R$ can be attributed to an increase in the population of active state conformers as well as a more robust coupling to the G protein.

## Allosteric network analysis reveals small negative allosteric modulation by cholesterol

Given the above observations, we employed rigidity-transmission allostery (RTA) analysis (*Sljoka, 2021*) to survey allosteric activation pathway perturbation by cholesterol within the ternary complex. The RTA algorithm is a computational tool based on mathematical rigidity theory and has been used to identify allosteric networks within proteins (*Jacobs et al., 2001*; *Sljoka, 2021*; *Whiteley, 2005*). It predicts how changes in the conformational rigidity or flexibility of one region in the protein are transmitted to distal sites by quantifying the resulting differences in the degrees of freedom within the system. Similarly, ligand-induced perturbations can be examined by rigidifying the ligand itself or its binding pocket. Using a model of an agonist- and GDP-bound $A_{2A}R$-$G_s\alpha\beta\gamma$ complex, equilibrated in a 1 μs simulation in POPC bilayer with 20% cholesterol, our previous work revealed that rigidification of the agonist NECA results in changes in the degrees of freedom which can be transmitted from the orthosteric pocket to the Gα nucleotide binding site (*Huang et al., 2021*). This allosteric network encompasses large portions of the receptor, the N- and C-terminal helices of Gα, parts of the Gα Ras domain, three out of seven beta propellers within Gβ, and a section of the Gβ N-terminal helix that forms coiled-coil interactions with Gγ.

Using the above model, we repeated the RTA analysis to examine whether this previously identified allosteric network is sensitive to the presence of cholesterol. Seven cholesterol molecules were found in the vicinity (within 6 Å) of $A_{2A}R$ and were removed prior to rigidification of the agonist. The resulting change in degrees of freedom is mapped in *Figure 5* for each residue within the ternary complex. Removal of cholesterol gave rise to an allosteric pathway which is very similar to that in the presence of cholesterol, although with altered intensities for some regions. Higher allosteric transmission is observed for the CWxP motif of TM6, in particular the tryptophan toggle switch W246[6.48], in the absence of cholesterol. On the other hand, stronger allosteric transmission is observed for the NPxxY motif of TM7 in the presence of cholesterol. Interestingly, the removal of cholesterol resulted in a slight overall enhancement in allosteric transmission to the G protein. This includes the Gα N- and C-terminal helices which interact with the receptor, as well as Gβ which has been found to play a role in conferring ligand efficacy (*Huang et al., 2021*). The above observations suggest that the overall presence of cholesterol, while not drastically perturbing, reduces signal transmission across the ternary complex. While these results are inconsistent with our experimental observations, it is also possible that there are indirect effects from cholesterol (e.g., stretching of the hydrophobic bilayer thickness) that override the effects predicted by the allosteric network analysis, as discussed below.

## $A_{2A}R$-cholesterol interactions are short-lived and non-specific

To further evaluate the nature of cholesterol-$A_{2A}R$ interactions, we carried out $^{19}F$ NMR experiments of fluorinated cholesterol analogs, delivered into either empty or $A_{2A}R$-embedded nanodiscs via MβCD. Two different molecules were tested (*Figure 6*). 3β-Fluoro-cholest-5-ene (3β-F-chol) was synthesized in house and features a fluorine atom in place of the cholesterol hydroxyl headgroup. The fluoro group is a relatively benign substitute for the hydroxyl due to its similar size and electronegativity. It also retains some ability to accept hydrogen bonds (*Hoffmann and Rychlewski, 2002*). Another cholesterol analog, referred to as F7-chol, was purchased commercially and had the tail isopropyl group replaced by $CF(CF_3)_2$. Incorporation of these analogs did not affect the response of $A_{2A}R$ toward ligands nor its ability to activate the G protein (*Figure 6—figure supplement 1*).

The NMR resonances were significantly broadened for both cholesterol analogs upon incorporation into the membrane (*Figure 6*). This is expected for lipid molecules situated in a slow-tumbling nanodisc. In the case of F7-chol, the peak shapes are further complicated by resonance overlap of the two $CF_3$ groups, which are inequivalent and exhibit complicated multiplicity patterns. Comparison between the spectra of 3β-F-chol in empty nanodiscs and $A_{2A}R$-embedded nanodiscs shows a clear environmental difference in the presence of receptor (*Figure 6A*). The resonance is ~0.5 ppm upfield shifted and broader relative to empty nanodiscs. However, the lack of difference in chemical shift or line width between 4% and 11% 3β-F-chol suggests that the above changes are predominantly a result

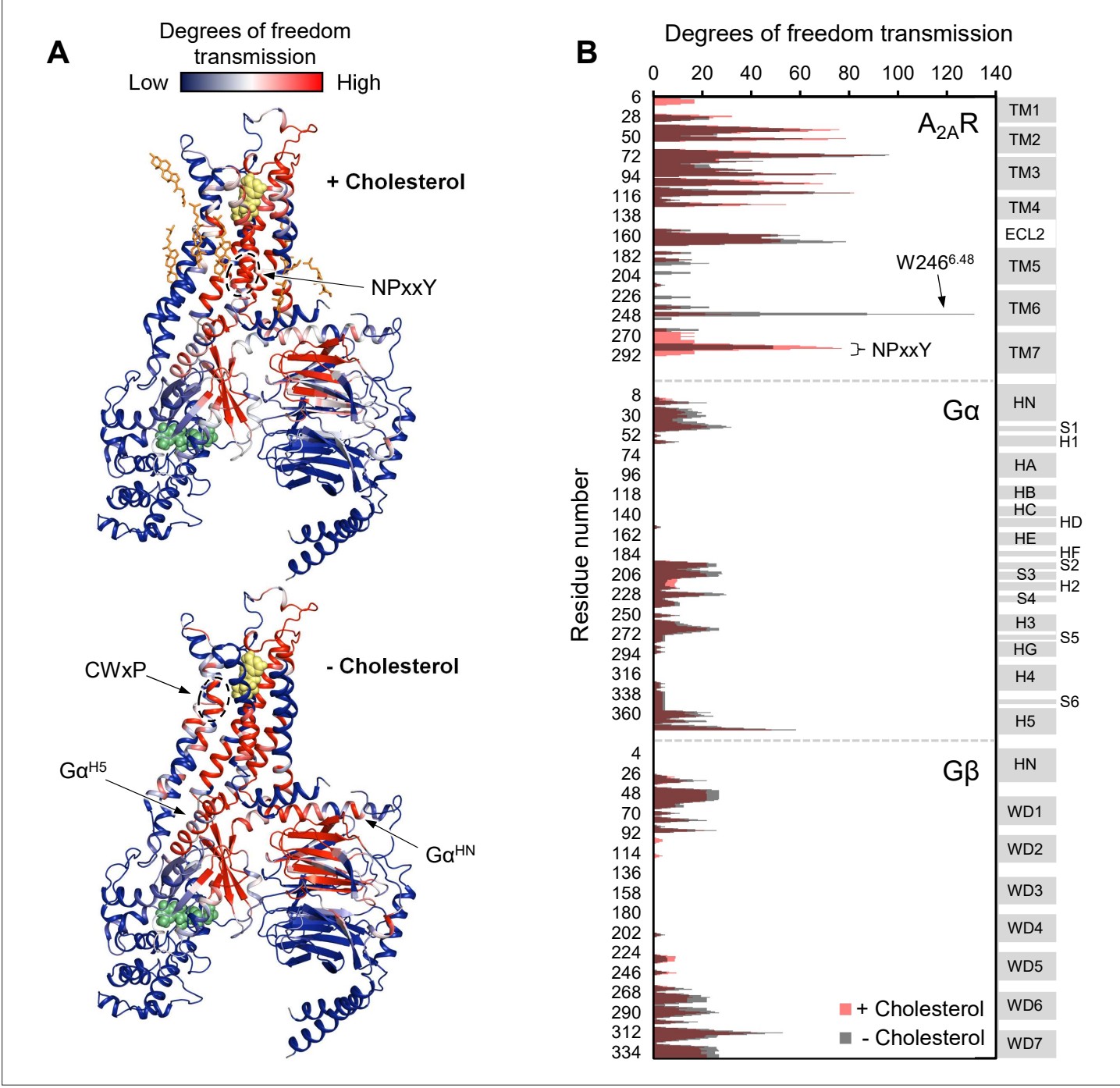

**Figure 5.** Removal of cholesterol leads to a small overall enhancement in allosteric transmission from the agonist binding site. (**A**) Allosteric networks within the $A_{2A}R$-$G\alpha\beta\gamma$ complex in the presence and absence of cholesterol, revealed through RTA analysis via rigidification of the agonist NECA (yellow spheres). The intensity of allosteric transmission is measured by the resulting regiospecific changes in degrees of freedom and is mapped in color (red/blue gradient bar). Cholesterol molecules are shown as orange sticks while green spheres represent GDP. (**B**) The intensity of allosteric transmission is plotted for each residue in $A_{2A}R$, $G\alpha$, and $G\beta$. Secondary structural elements are indicated on the right. Gray blocks denote $\alpha$-helices and $\beta$-strands, while white gaps represent loops. For the $G\alpha$ subunit, the common $G\alpha$ numbering system is used (*Flock et al., 2015*).

of altered environment (i.e., availability of hydrophobic proteinaceous surfaces) rather than a shift toward receptor-bound states at specific binding pockets.

The spectra of F7-chol are harder to interpret. Due to the two overlapping $CF_3$ resonances, a small change in chemical shift for either one could bring about dramatic variation in the overall peak shape

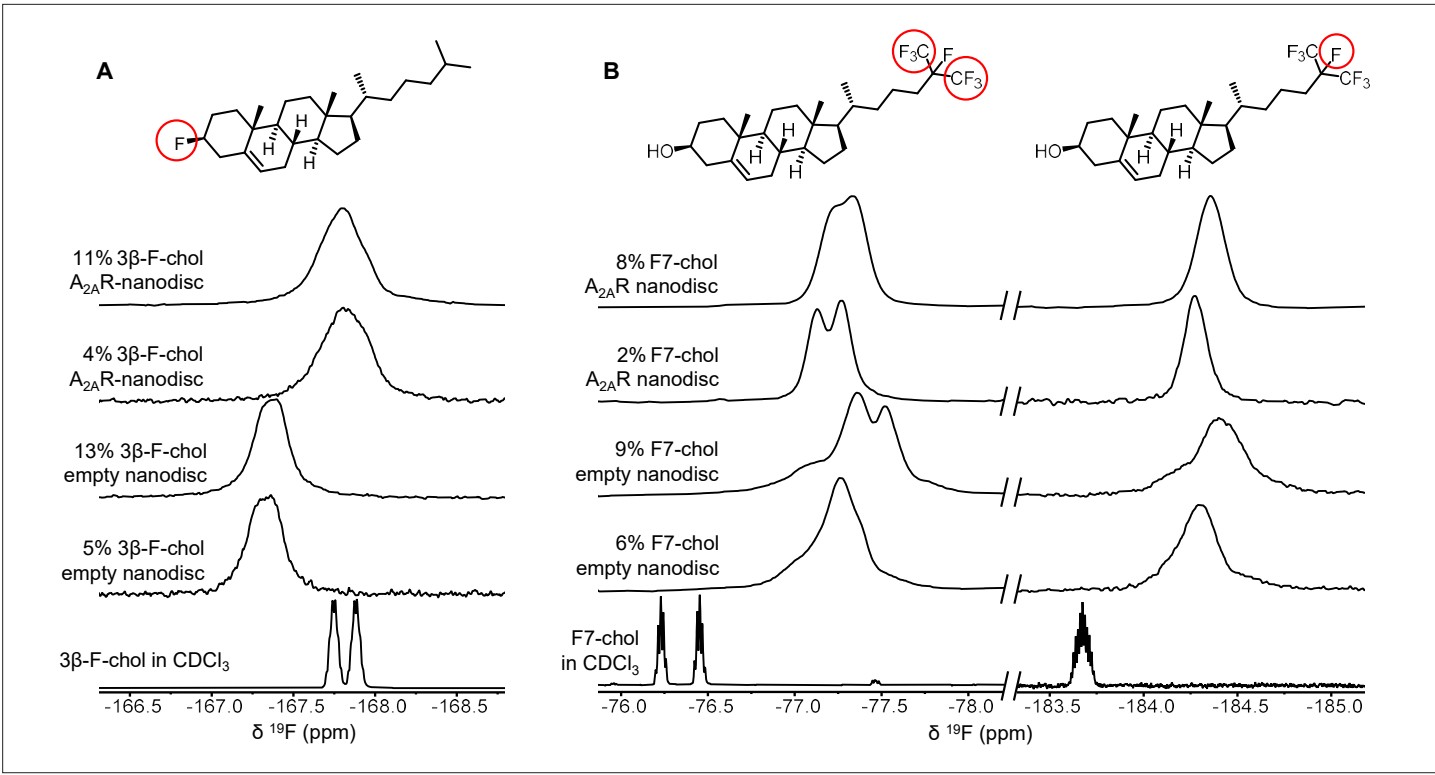

**Figure 6.** $^{19}$F-cholesterol analogs interact with A$_{2A}$R but show no clear evidence of a long-lived bound state. Non-decoupled $^{19}$F NMR spectra of 3β-F-chol (**A**) and F7-chol (**B**) in chloroform, empty nanodisc, and A$_{2A}$R (apo)-embedded nanodisc. The fluorine groups contributing to each of the resonances are circled and shown above the corresponding peak.

The online version of this article includes the following figure supplement(s) for figure 6:

**Figure supplement 1.** Incorporation of $^{19}$F-cholesterol analogs into nanodiscs did not affect A$_{2A}$R ligand sensitivity and G-protein activation.

**Figure supplement 2.** $^{19}$F-cholesterol analogs behave similarly in the presence of agonist- and inverse agonist-bound A$_{2A}$R.

(*Figure 6B*). For example, it is clear from the two empty nanodisc spectra (containing either 6% or 9% F7-chol) that the membrane environment is altered with increasing F7-chol. Therefore, we cannot be confident about whether the observed changes in the CF$_3$ peaks in the presence of to A$_{2A}$R are a consequence of specific binding or simply changes in the membrane environment. Based on the rough chemical shift values of the CF$_3$ and CF resonances, the latter explanation is more probable. Overall, the NMR data from the two $^{19}$F-cholesterol analogs show environmental differences between empty and A$_{2A}$R-embedded nanodiscs as well as between different cholesterol concentrations. However, there is no direct evidence of a long-lived receptor-bound state.

In the case of a classical PAM, stronger receptor binding is expected in the presence of an agonist or G protein, versus an inverse agonist. The opposite would hold for a classical negative allosteric modulator (NAM). Yet, there was no apparent difference in chemical shift sensitivity toward agonist or inverse agonist for either $^{19}$F cholesterol analogs (*Figure 6—figure supplement 2*). The NMR spectra of 3β-F-chol in A$_{2A}$R-embedded nanodiscs are nearly identical upon the addition of inverse agonist, full agonist, and mini-G$_s$, a G-protein mimetic that has been shown to stabilize the A$_1$ active state (*Carpenter et al., 2016*; *Huang et al., 2021*). Small chemical shift changes were observed for F7-chol between the apo receptor and the ligand/mini-G$_s$-bound conditions. However, the direction of shift is the same between full agonist and inverse agonist. Thus, cholesterol interactions are independent from the identity of the orthosteric ligand bound to A$_{2A}$R, despite being observed as a functional PAM in vitro (*Figures 2–3*) and predicted as a NAM in silico (*Figure 5*). It is distinctly possible that the small shift perturbations observed with ligand and mini-G$_s$ are a consequence of F7-chol contacting multiple sites on the receptor. Nevertheless, the lack of any pronounced shift perturbation (particularly in the presence of agonist since we observe cholesterol to behave as a PAM) leads us to consider that cholesterol interactions are transient. It is therefore more likely that the observed positive allosteric

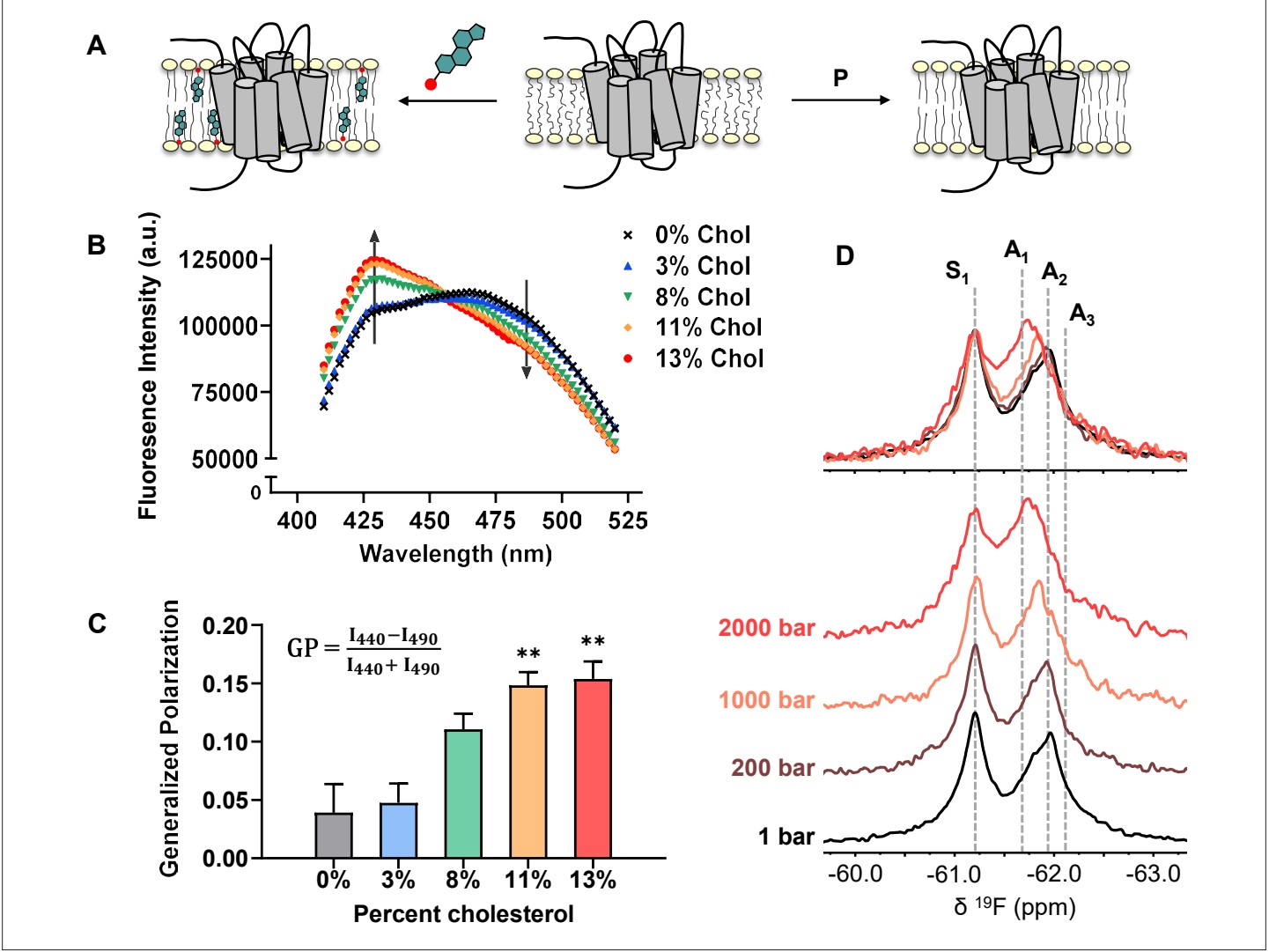

**Figure 7.** Cholesterol allostery in $A_{2A}R$ is likely a result of indirect membrane effects. (**A**) The lipid bilayer can be rigidified and thickened upon addition of cholesterol or an increase in lateral pressure. (**B**) Averaged fluorescence spectra (n = 4) of Laurdan in $A_{2A}R$-embedded nanodiscs containing varying concentrations of cholesterol. (**C**) The emission intensity of Laurdan at 440 nm and 490 nm were used to calculate the generalized polarization values. Data represent mean ± SEM (n = 4, technical triplicates). Astrisks represent statistical significance over both the 0% and the 3% conditions. Statistical significance was determined via one-way ANOVA followed by Tukey's multiple comparisons test. ** $p \leq 0.01$. (**D**) $^{19}F$ NMR spectra of $A_{2A}R$, in the absence of ligand or cholesterol, acquired at 1, 200, 1000, and 2000 bar pressures. The key functional states are indicated by gray dashed lines.

effects of cholesterol are predominantly indirect and relayed through the physical changes to the membrane bilayer.

## Cholesterol allostery in $A_{2A}R$ may be a result of indirect membrane effects

The effects of cholesterol on the physical properties of lipid bilayers have been well documented. The planar structure of cholesterol promotes orientational order in the liquid disordered phase of the bilayer, leading to reduced lateral diffusion and increased hydrophobic thickness (*Figure 7A*; *Crane and Tamm, 2004*; *de Meyer and Smit, 2009*; *Filippov et al., 2003*; *Hung et al., 2007*). For instance, as much as 20% increase in thickness can be expected for a POPC bilayer when the cholesterol concentration is varied from 0% to 30% (*Mouritsen and Bagatolli, 2016*; *Tharad et al., 2018*).

We employed the lipophilic fluorescent probe Laurdan to monitor the membrane orientational order of $A_{2A}R$-embedded nanodiscs as a function of cholesterol. The generalized polarization (GP = $\frac{I_{440} - I_{490}}{I_{440} + I_{490}}$) of Laurdan fluorescence is a consequence of solvent contact. Fluid (disordered) membranes

exhibit smaller GP values, a consequence of dipole relaxation between Laurdan and nearby water molecules which causes a red shift of the emission wavelength. In a phospholipid bilayer, greater water accessibility at the hydrophobic-hydrophilic interface is typically a signature of enhanced reorientational dynamics within the lipid milieu (*Yu et al., 1996*). As shown in *Figure 7B–C*, we observed a consistent shift of the Laurdan fluorescence spectra which gave rise to higher GP values at elevated cholesterol concentrations. This indicates that cholesterol incorporation enhances lipid orientational order in $A_{2A}R$-embedded nanodisc bilayers.

An enhanced orientational order arises from a higher fraction of trans conformers in the lipid chains and consequently, an increased hydrophobic thickness. In the case where $A_{2A}R$ adopts an ensemble of states, the equilibrium is expected to shift toward those states which are more compatible with an increased hydrophobic thickness (*Andersen and Koeppe, 2007*). Bilayer thickness can be readily modulated by changing the composition of lipids or acyl chain lengths. Alternatively, the application of hydrostatic pressure can be used in an NMR experiment to affect changes in bilayer properties while avoiding potential complications associated with specific lipid-receptor interactions. For a typical liquid-crystalline phosphatidylcholine bilayer, pressure-induced compression is far more significant in the lateral than in the transverse/perpendicular direction (*Stamatoff et al., 1978*). An elevation in pressure at constant temperature promotes ordering of the fatty acyl chains. This leads to an increase in lipid packing density and hydrophobic thickness, and a reduction of lateral diffusion (*Ding et al., 2017*). Thus, hydrostatic pressure provides an effective way to mimic the ordering effects of cholesterol on a lipid bilayer (*Figure 7A*).

We recorded the $^{19}F$ NMR spectra of the apo receptor in nanodiscs without cholesterol, at a pressure of 1, 200, 1000, and 2000 bar. Like increasing cholesterol, the rise in pressure resulted in a bias toward the active ensemble, particularly the $A_1$ (full agonist) state (*Figure 7D*). Interestingly, the magnitude of change is non-linear and considerably larger at 2000 bar in comparison to 1000 and 200 bar. This is consistent with the expected change in membrane thickness as a function of pressure. For a pure POPC bilayer at 20 °C, the increase in hydrophobic thickness is small (up to ~2 Å) and roughly linear below 1200 bar. Above this pressure, the bilayer transitions to a solid ordered phase which results in a rapid increase of membrane thickness on the order of 10 Å (*Rappolt et al., 2003*). In comparison, the thickness increase as a result of 10–15% cholesterol is on the order of 1–3 Å (*Hung et al., 2007*).

The NMR results are similar to that of previous pressure studies of the $\beta_1$AR and $\beta_2$AR in detergent micelles (*Abiko et al., 2019*; *Lerch et al., 2020*). In both cases, a shift toward the active state was observed in response to pressure, which was correlated with a reduction in void volume of the active receptor relative to the inactive form. Here, $^{19}F$ NMR allowed a more detailed delineation of the conformational landscape of $A_{2A}R$. Unlike agonist- or G-protein-induced activation, where the inactive ensemble is significantly diminished and all three active state conformers are promoted, the redistribution of states brought about by pressure saw a smaller decrease of the inactive ensemble and a specific shift in equilibrium toward the $A_1$ state (*Figure 7D*). The effects from pressure directly exerted on the receptor cannot be easily separated from indirect effects that are manifested through changes in the lipid bilayer. However, more influence from the membrane is expected since the molecular assembly of lipid bilayers is much more sensitive to pressure relative to the conformation of proteins (*Kato and Hayashi, 1999*). Both lipid bilayers and detergent micelles are well-known soft systems whose hydrophobic dimensions change precipitously with pressure and whose compressibilities are thus significantly higher than those of membrane proteins (*Alvares et al., 2014*; *Lerch et al., 2020*). The lipid bilayer, relative to detergent micelles, was shown to protect integral membrane proteins from pressure-induced denaturation (*Kangur et al., 2008*). Overall, our pressure-resolved NMR data suggest that $A_{2A}R$ can be regulated indirectly through changes in the lipid bilayer. While the mechanism may be complex and the effects are subtle, receptor activation appears to be favored in an environment with higher packing density, acyl chain order, and hydrophobic thickness.

## Discussion

$A_{2A}R$ has been intensely studied by both X-ray crystallography and more recently by electron cryomicroscopy (cryo-EM). In many cases, cholesterol or CHS have proven useful in stabilizing the receptor and obtaining high-resolution structures. Earlier in vitro and cell-based studies, along with the clear delineation of cholesterol in many crystal structures of $A_{2A}R$ suggest that the molecule may play a

direct allosteric role in modulating receptor function. A body of computational work has since show-cased cholesterol hot spots across the receptor and some of these studies proposed state-dependent interactions (*Lovera et al., 2019*; *McGraw et al., 2019*). Nevertheless, there is no literature consensus on the allosteric role of cholesterol on this prototypical GPCR.

In this study, we set out to investigate both the magnitude and origin of the allosteric interplay between cholesterol and $A_{2A}R$ in phospholipid bilayers, using an identical model membrane system for all functional assays and biophysical experiments. Nanodiscs have been used extensively in functional and structural characterization of complex membrane proteins (*Sligar and Denisov, 2021*). The protocols used in the current experiments generated monodisperse 8 nm diameter nanodiscs containing a single receptor and roughly 35–40 lipids per leaflet in addition to cholesterol (*Hagn et al., 2013*; *Huang et al., 2021*). This is a reductionist system featuring a single receptor surrounded by a fluid POPC/POPG lipid bilayer, with 1–5 cholesterol molecules per leaflet across the concentration range that was investigated. Thus, cholesterol-mediated lateral phase separation, or receptor oligomerization are excluded in this analysis. On the other hand, the receptor can be complexed with heterotrimeric G protein, and the role of cholesterol in modulating the receptor's state distribution and G-protein coupling can be studied with exquisite sensitivity.

Here, functional and spectroscopic studies in nanodiscs identify cholesterol as a weak PAM. Specifically, GTP hydrolysis assays found a marginal increase in basal activity with increasing cholesterol, in addition to a weak enhancement in the agonist potency. $^{19}F$ NMR experiments revealed little or no difference in the receptor spectra upon addition of 4% cholesterol. A very modest shift in equilibrium toward the active states ($A_1$ and $A_2$) was observed at 13% cholesterol, corroborating the observed allosteric effects. A distinct enhancement of $A_3$ is also found at 13% cholesterol for the apo receptor bound to G protein, implying that cholesterol either directly or indirectly stabilizes the precoupled $A_{2A}R$-$G\alpha\beta\gamma$ complex.

Despite the observed enhancement in both G-protein coupling and the activation states, $^{19}F$ NMR of two fluorinated cholesterol analogs implied a weak or transient interaction between cholesterol and $A_{2A}R$. Moreover, the confining conditions of the nanodisc would be expected to favor a cholesterol-bound state. There was also no correlation between the chemical shifts of the cholesterol analogs and orthosteric ligand efficacy, suggesting that the origin of the observed positive allostery is through the indirect effects of cholesterol on the membrane itself.

Laurdan fluorescence experiments confirmed that lipid orientational order is increased by cholesterol in the presence of $A_{2A}R$. Similarly, hydrostatic pressure is well-known to give rise to increased orientational order in lipid bilayers (*Ding et al., 2017*; *Nicolini et al., 2006*). We therefore sought to use hydrostatic pressure as a surrogate to cholesterol to consider potential indirect effects on receptor function that arise simply from membrane ordering. Under conditions where hydrostatic pressure is known to exert similar changes in membrane thickness as 11–13% cholesterol, $^{19}F$ NMR revealed comparable responses in the conformational ensemble of the receptor – namely a shift in equilibrium to the active states at pressures of 1000 bar or more. We note that denaturation of the protein with these pressures was not observed, based on the observation of $^{19}F$ NMR spectra of misfolded $A_{2A}R$ which tend to exhibit a single upfield resonance whose shift is insensitive to ligand (unpublished results).

While we cannot rule out the possibility of a cumulative influence from multiple fast-exchanging, weakly binding interactions, results from the current study strongly suggest that changes in membrane physical properties are the primary means by which cholesterol regulates $A_{2A}R$. There may also be subtle NAM effects from direct interaction with cholesterol, as suggested by our computational analysis, which at the same time are overcome by stronger indirect effects through the membrane. It is possible that this is also the mechanism through which CHS enhances the ligand binding activity of $A_{2A}R$ in detergent micelles (i.e., by modulating the micellar structure to a more bilayer-like morphology). In support of this idea, an evaluation of $A_{2A}R$ reconstituted in various mixed micelle systems revealed a correlation between receptor activity to those detergent/CHS compositions that gave rise to a micellar hydrophobic thickness that closely matches that of native mammalian bilayers (*O'Malley et al., 2011b*).

Although the nanodisc is an effective membrane mimetic, it is not a perfect replacement of the biological membrane. For example, lipids in nanodiscs exhibit slightly more motional freedom in their headgroup region and less motional freedom at the center of the bilayer in comparison to liposomes

(*Stepien et al., 2015*). They also display a broader phase transition in response to temperature in addition to a transition temperature that is 3-4 °C higher than that of liposomes (*Shaw et al., 2004*). Nevertheless, we have not performed detailed pressure-dependent NMR studies of the POPC/POPG lipids to confirm that a bilayer-like morphology is maintained throughout the range of pressures used in these experiments.

Another limitation of the current study is the range of cholesterol concentrations being probed, which is below the physiological norm. In cell-based experiments, total cholesterol depletion is not possible without adversely affecting cellular integrity. In many cases, the amount of cholesterol left in the membrane was not quantified and the focus was instead on the disruption of raft-like domains. Our nanodisc samples contained 0–13% cholesterol, which is below the concentration regime for raft formation (*Barrett et al., 2013*; *Crane and Tamm, 2004*). These two strategies (extraction from cholesterol-rich membranes and delivery into cholesterol-free membranes) explore largely different processes; the former involves the disruption of rafts while the latter allows studies of the interaction of cholesterol species with the receptor in a monomeric state.

Our data suggests that such interactions, if present for $A_{2A}R$, are non-specific and short-lived. This may explain why structural and computational work has yet to converge upon a single cholesterol binding site. Like lipids, the observation of cholesterol in crystal structures may simply be a consequence of having cholesterol as a part of the crystallization matrix. In fact, many $A_{2A}R$ structures which do not contain co-crystallized cholesterol (all the active state structures and some inactive state structures) had the molecule present in large quantities during crystallization. In one example, complexes of $A_{2A}R$ bound to an engineered mini-G protein were crystallized in octylthioglucoside micelles either in the presence or absence of CHS. No discernible difference was found between crystals that grew with or without CHS and the structure was solved using data collected from two crystals, one from each condition (*Carpenter et al., 2016*). Similarly, the numerous cholesterol 'hot spots' predicted through computational approaches may not necessarily indicate functional specificity, but rather geometric compatibility between certain hydrophobic patches or grooves surrounding the receptor and the cholesterol backbone. This is reflected in the fact that nearly all seven transmembrane helices and grooves between helices in $A_{2A}R$ have been predicted in various studies to bind cholesterol (*Genheden et al., 2017*; *Guixà-González et al., 2017*; *Lee and Lyman, 2012*; *Lovera et al., 2019*; *McGraw et al., 2019*; *Rouviere et al., 2017*; *Sejdiu and Tieleman, 2020*; *Song et al., 2019*). Furthermore, the presence of CCM or CRAC motifs has recently been shown to not be predictive of cholesterol binding in GPCRs (*Taghon et al., 2021*).

The current work shows that $A_{2A}R$ does not require cholesterol to function in an in vitro bilayer setting. However, many experiments have highlighted the role of cholesterol-rich domains for $A_{2A}R$ to function in a cellular context. As alluded to above, a major shortcoming of our nanodisc system is the upper limit of cholesterol that can be delivered. This prevented us from evaluating the system at higher, more physiological cholesterol concentrations and probing the effects from protein partitioning between liquid ordered and liquid disordered phases (*Gutierrez et al., 2019*). It is unclear whether $A_{2A}R$ alone prefers certain regions on the plasma membrane. Nevertheless, both the stimulatory G protein and many isoforms of adenylyl cyclase were shown to partition into raft-like domains (*Kamata et al., 2008*; *Oh and Schnitzer, 2001*; *Ostrom et al., 2001*; *Ostrom and Insel, 2004*). Spatial co-localization of the receptor with other cellular binding partners in these membrane regions may therefore be required to form and maintain signaling complexes.

## Materials and methods

**Key resources table**

| Reagent type (species) or resource | Designation | Source or reference | Identifiers | Additional information |
|---|---|---|---|---|
| Strain, strain background (*Pichia pastoris*) | SMD 1163 | Invitrogen | | $A_{2A}R$ expression host |
| Strain, strain background (*Escherichia coli*) | BL21 (DE3) | Invitrogen | Cat#: C600003 | Gα expression host |

*Continued on next page*

*Continued*

| Reagent type (species) or resource | Designation | Source or reference | Identifiers | Additional information |
|---|---|---|---|---|
| Strain, strain background (*Spodoptera frugiperda*) | Sf9 | ATCC | ATCC: CRL-1711 | Gβγ expression host |
| Recombinant DNA reagent | Plasmid: pET15b containing wild type $G_s\alpha$ | This paper | | The plasmid is available upon request to the corresponding author |
| Commercial assay or kit | GTPase-Glo | Promega | Cat#: V7681 | For GTP hydrolysis assay |
| Commercial assay or kit | Cholesterol quantification kit | R-Biopharm and Roche Diagnostics | Cat#: 10139050035 | |
| Chemical compound, drug | 2-Bromo-N-(4-(trifluoromethyl)phenyl)acetamide (BTFMA) | Apollo Scientific | Cat#: PC8478 | |
| Chemical compound, drug | 5'-N-Ethylcarboxamidoadenosine (NECA) | Tocris | Cat#: 1,691 | |
| Chemical compound, drug | LUF 5834 | Tocris | Cat#: 4,603 | |
| Chemical compound, drug | ZM 241385 | Tocris | Cat#: 1,036 | |
| Chemical compound, drug | Methyl-β-cyclodextrin | Millipore Sigma | Cat#: 332,615 | |
| Chemical compound, drug | Guanosine 5'-diphosphate (GDP) | Millipore Sigma | Cat#: 51,060 | |
| Chemical compound, drug | 1-Palmitoyl-2-oleoyl-glycero-3-phosphocholine (POPC) | Avanti Polar Lipids | Cat#: 850457 C | |
| Chemical compound, drug | 1-Palmitoyl-2-oleoyl-sn-glycero-3-phospho-(1'-rac-glycerol) (POPG) | Avanti Polar Lipids | Cat#: 840457 C | |
| Chemical compound, drug | Cholesterol | Millipore Sigma | Cat#: C8667 | |
| Chemical compound, drug | F7-Cholesterol (25,26,26,26,27,27,27-heptafluorocholesterol) | Avanti Polar Lipids | Cat#: 700,002 | |
| Chemical compound, drug | 3β-Fluoro-cholest-5-ene | This paper | | Synthetic methods are described in this paper. The compound is available upon request to the corresponding author |
| Chemical compound, drug | Laurdan | Millipore Sigma | Cat#: 850582 P | |
| Software, algorithm | MestReNova version 12.0.4 or higher | Mestrelab Research | https://mestrelab.com/ | |

## $A_{2A}R$ expression, purification, and nanodisc reconstitution

Receptor cloning, expression, and purification have been described previously (*Huang et al., 2021*; *Ye et al., 2016*). Briefly, *Pichia pastoris* (*P. pastoris*) SMD 1163 (*Δhis4 Δpep4 Δprb1*) cells carrying the gene for $A_{2A}R$ (residues 2–317 with the V229C mutation for $^{19}F$-labeling) were grown to high density in either shaker flasks or a bioreactor. Methanol (5% v/v) was added every 12–16 hr to induce expression and the cells were harvested after 60–72 hr post induction. The receptors were extracted from the yeast membrane, reacted with the fluorine tag 2-Bromo-N-[4-(trifluoromethyl)phenyl] acetamide (BTFMA) when applicable, and further purified in the absence of cholesterol or cholesterol analogs. Prior to cholesterol incorporation, the receptors were reconstituted in rHDL nanodiscs using a 3:2 ratio of 1-palmitoyl-2-oleoyl-sn-glycero-3-phosphocholine (POPC) to 1-palmitoyl-2-oleoyl-sn-glycero-3-phospho-(1'-rac-glycerol) (POPG) and the MSPΔH5 membrane scaffold protein (*Hagn et al., 2013*). The sample was purified using a HiLoad 16/600 Superdex 200 preparatory grade size exclusion column equilibrated with nanodisc storage buffer (50 mM HEPES, pH 7.4, 100 mM NaCl), and the peak containing monodisperse nanodiscs were collected for cholesterol incorporation and further purification.

## Incorporation of cholesterol and cholesterol analogs

Incorporation of cholesterol and its fluorinated analogs in nanodiscs was achieved via incubation of the nanodiscs with cholesterol solubilized in methyl-β-cyclodextrin (MβCD, MilliporeSigma Canada, Oakville, Canada). One to 2 days prior to incorporation, a concentrated MβCD-cholesterol stock was prepared by mixing cholesterol (MilliporeSigma) with MβCD buffer (50 mM HEPES, pH 7.4, 100 mM NaCl, 40 mM MβCD) to a final concentration of 8 mM (4 mM in the case of fluorinated analogs, due to their increased hydrophobicity). The mixture was sonicated briefly to disperse any large chunks, then incubated at 30 °C for 24–36 hr with shaking until the solution is clear to the eye. The solution is filtered through a 0.2 µM filter to eliminate any undissolved particles, then diluted with MβCD buffer to make MβCD-cholesterol stocks containing 0.8 mM, 2 mM, 3 mM, and 4 mM cholesterol (0.8 mM and 3 mM in the case of fluorinated analogs). These stocks were mixed with nanodiscs collected from the size exclusion column described above (containing both empty and $A_{2A}R$-embedded nanodiscs, which co-eluted) in a 1:3 v/v ratio, such that the final concentrations in the mixtures are 10 mM MβCD, 20–30 µM nanodisc, and 0.2 mM, 0.5 mM, 0.75 mM, or 1 mM cholesterol, for different levels of cholesterol incorporation. The mixtures were incubated at room temperature for 15 min with gentle shaking, then diluted 10-fold with nanodisc storage buffer containing 1–2 mL bed volume of Ni-NTA resin prior to incubation at 4 °C for 2 hr. After incubation, Ni-NTA resins were collected using a gravity column and washed extensively with nanodisc storage buffer to remove residual empty nanodiscs, MβCD, and MβCD-cholesterol. Nanodiscs containing the His-tagged $A_{2A}R$ were eluted from the column using elution buffer (50 mM HEPES, pH 7.4, 100 mM NaCl, 250 mM imidazole), concentrated, and exchanged to nanodisc storage buffer for subsequent experiments. For empty nanodiscs, a $His_6$-tagged MSPΔH5 protein was used. The reconstituted discs were treated with MβCD-cholesterol as above, incubated with Ni-NTA resins, and the MβCD was washed away prior to elution and concentration.

## Lipid quantification

Phospholipid concentrations were measured using a modified sulfo-phospho-vanillin assay (***Frings and Dunn, 1970***). Each sample containing unsaturated phospholipids (nanodiscs or phospholipid standards) was dissolved in 50-fold volume of concentrated sulfuric acid and incubated in a boiling water bath for 10 min. The samples were cooled in a cold-water bath for 5 min, then diluted 16-fold with a phospho-vanillin reagent (0.12% w/v vanillin dissolved in 68% v/v phosphoric acid). The samples were incubated in the dark for 30 min prior to absorbance measurements at 525 nm using a spectrophotometer. Lipid concentrations were determined using standard curves of $A_{525}$ from pure POPC and POPG. In the case of nanodiscs, the lipid concentrations were determined using standard curves of both POPC and POPG:

$$[Lipid_{nanodisc,adjusted}] = \tfrac{3}{5}[Lipid_{nanodisc,POPC}] + \tfrac{2}{5}[Lipid_{nanodisc,POPG}]$$

## Quantification of cholesterol and fluorinated cholesterol analogs

Cholesterol concentrations were measured calorimetrically using a commercial kit (R-Biopharm and Roche Diagnostics, Cat. No. 10139050035) following the manufacturer's protocol. The concentrations of 3β-F-cholesterol and F7-cholesterol (Avanti Polar Lipids) were estimated via integration of $^{19}F$ NMR resonances of the cholesterol analog in relation to a reference compound (fluoroacetate in the case of 3β-F-cholesterol and trifluoroacetate in the case of F7-cholesterol), where the relative signal loss in the reference peak due to shortened relaxation delay was corrected for. Percent cholesterol in a given sample was calculated as follows:

$$\% \, cholesterol = \frac{[Cholesterol]}{[Lipid]} \times 100$$

## G-protein cloning, expression, and purification

The expression and purification of $G_s\alpha$, $G\beta\gamma$, and mini-$G_s\alpha$ have been described previously (***Huang et al., 2021***) with the only difference being that a wild-type $G_s\alpha$ was used in the current work. To generate this construct, a double-stranded DNA fragment for the wild-type $G_s\alpha$ short isoform was codon optimized and synthesized using the GeneArt service from Thermofisher. This fragment carried overlapping sequences with the previously described pET15b MBP-$G_s\alpha$ mutant sequence (***Huang***

*et al., 2021*). The plasmid was digested with XhoI and SacI (New England BioLabs, Ipswich, MA, USA) to remove the mutant $G_s\alpha$ sequence and purified via electrophoresis and gel extraction kit (Bio Basic, Markham, Canada). The resulted plasmid backbone and DNA fragment were fused using the pEasy assembly kit from TransGen Biotech following manufacturer's instructions. The plasmid was transformed into *Escherichia coli* (*E. coli*) BL21 (DE3) cells and a resulting colony containing the gene for the wild-type $G_s\alpha$ was selected for protein expression.

## NMR experiments

NMR samples were prepared in nanodisc storage buffer with 20–100 µM $A_{2A}R$-V229C (BTFMA-labelled for receptor NMR, unlabeled for $^{19}F$-cholesterol NMR), 10% $D_2O$, and 20 µM sodium trifluoroacetate (TFA, for receptor NMR) or 100 µM fluoroacetate (for $^{19}F$-cholesterol NMR) as the $^{19}F$ chemical shift reference. For samples containing G protein (1.1-fold excess), the buffer also included 100 µM GDP, 2 mM $MgCl_2$, and 5% glycerol. When applicable, $A_{2A}R$ ligands were added at saturating concentrations (1 mM NECA, 500 µM LUF5834, or 500 µM ZM241385). All samples were sterile-filtered and prepared in sterile Shigemi tubes to prevent microbial contamination. NMR experiments were acquired at 20 °C on a 500 MHz Varian Inova spectrometer equipped with a 5 mm room temperature inverse HFX probe. A typical fluorine NMR experiment included a 100ms recycle delay, a 5.5 µs (45°) excitation pulse, and a 500ms acquisition time. Each experiment acquired between 100,000–400,000 scans, yielding a S/N of approximately 50–100. Spectra were processed using MestReNova (Mestrelab Research S.L.) employing chemical shift referencing (–75.6 ppm for TFA and –217 ppm for fluoroacetate), baseline correction, zero filling, and exponential apodization equivalent to a 5–20 Hz line broadening. For high-pressure NMR, the sample was transferred to a 3 mm zirconia tube (Daedalus Innovations, Aston, PA, USA) and covered with paraffin oil. The tube was placed inside a 600 MHz Varian Inova spectrometer equipped with a triple-resonance cryoprobe tunable to $^{19}F$, via a stainless-steel fluid line connected to an Xtreme-60 syringe pump (Daedalus Innovations) prefilled with paraffin oil as the pressurizing fluid. Pressure was increased at a rate of 100 bar/min to the desired value, and the sample was equilibrated for 5 min at the final set pressure prior to acquisition at 20 °C.

## Membrane fluidity measurements

$A_{2A}R$-embedded nanodiscs were incubated with the fluorescent probe Laurdan (MilliporeSigma) at room temperature for 30 min in the dark at a final concentration of 1 µM $A_{2A}R$ and 10 µM Laurdan (diluted from a 10 mM dimethylformamide stock). Free Laurdan was removed by extensive buffer-exchange with the nanodisc storage buffer and subsequently filtering the sample through a 0.2 µm filter. Flow-through from the final round of buffer-exchange was kept for background correction. The samples were transferred to a black 384-well plate and the fluorescent emission spectra (410 nm – 520 nm) were acquired using a TECAN Spark multi-mode plate reader (Tecan, Männedorf, Switzerland) at 26 °C with an excitation wavelength of 350 nm. Each emission spectrum was background-corrected, then area-normalized to the 13% cholesterol condition. The generalized polarization (GP) of each sample was determined using the formula $GP = \frac{I_{440}-I_{490}}{I_{440}+I_{490}}$ , where $I_{440}$ and $I_{490}$ represent the emission intensities at 440 nm and 490 nm, respectively.

## GTP hydrolysis experiments

GTP hydrolysis experiments were carried out using the GTPase-Glo assay kit (Promega, Madison, WI, USA) following the manufacturer's protocol (*Mondal et al., 2015*). Briefly, purified receptor and G protein were incubated at room temperature in a buffer containing 50 mM HEPES, pH 7.4, 100 mM NaCl, 2 mM $MgCl_2$, 1 µM GDP, and 4 µM GTP, at a final concentration of 250 nM G protein, 250 nM $A_{2A}R$, and various concentrations of the agonist NECA. Control reactions included buffer with GTP but in the absence of either $A_{2A}R$ or both $A_{2A}R$ and G protein. After 90 min, unreacted GTP was converted to ATP prior to the addition of a detection reagent containing luciferase. The resulting luminescence, which is proportional to the amount of unreacted GTP, was measured using a TECAN Spark multi-mode plate reader with an integration time of 1 min. GTP hydrolysis was determined as follows:

G protein only (in the absence of $A_{2A}R$):

$$\Delta Lum_G = Lum\left(bufferonly\right) - Lum\left(Gproteinonly\right)$$

In the presence of $A_{2A}R$:

$$\Delta Lum_{G+R} = Lum\left(bufferonly\right) - Lum\left(Gprotein + A_{2A}R\right)$$

where Lum is the luminescence signal intensity.

The relative GTP hydrolysis for each $A_{2A}R$ (NECA) sample was calculated as follows:

$$RelativeGTPhydrolysis = \frac{\Delta Lum_{G+R}}{\Delta Lum_G} \times 100$$

The NECA dose-response data were fit using a variable slope model in GraphPad Prism 8.4.2 employing the equation:

$$Response = E_{min} + \frac{(x^n)\left(E_{max} - E_{min}\right)}{x^n + \left(EC_{50}\right)^n}$$

where $x$ is the agonist concentration, $E_{min}$ is the minimum response, $E_{max}$ is the maximum response, $EC_{50}$ is the agonist concentration that promotes half-maximum response, and $n$ is the Hill coefficient.

## Dynamic light scattering

DLS samples were prepared in nanodisc storage buffer containing 5 µM $A_{2A}R$-embedded nanodiscs supplemented with different mol% of cholesterol. Each sample was filtered through a 0.2 µm syringe filter to remove large dust particles before transferring to a small-volume 10 mm quartz cuvette (Starna Cells, Atascadera, CA, USA). DLS measurements were carried out inside a Zetasizer Nano-ZS particle size analyzer (Malvern Panalytical, Malvern, United Kindom) equipped with a He-Ne laser ($\lambda$ = 633 nm). Samples were equilibrated at 25 °C for 2 min and the scattered light was measured at a 173° backscatter angle. The resulting correlation function was analyzed using the general purpose (non-negative least squares) analysis model in the Zetasizer software (v7.13, Malvern Panalytical) for distribution analysis, assuming a buffer viscosity of 0.9066 cP, a buffer refractive index of 1.332, and a protein refractive index of 1.450. Data was averaged over three or four independent trials, each having three replicate measurements of 10–20 scans.

## Synthesis of 3β-fluoro-cholest-5-ene

3β-Fluoro-cholest-5-ene was synthesized from cholesterol in one step, using the deoxyfluorination reagent DAST (diethylaminosulfur trifluoride, Toronto Research Chemicals, North York, Canada). Although fluorinations with DAST often proceed through an $S_N2$ mechanism, fluorination of cholesterol is known to retain its stereochemistry (*Rozen et al., 1979*). This results from homoallylic participation forming a carbonium ion intermediate (*Li et al., 2016*; *Rozen et al., 1979*).

Cholesterol (650 mg, 1.68 mmol) was dissolved in dry $CH_2Cl_2$ (15 mL) in a plastic reaction vessel under argon. The mixture was cooled to –20 °C and DAST (four eq., 0.89 mL) was added dropwise over 5 min. The solution was stirred at –20 °C for 1 hr. The cooling bath was removed, and the reaction was continued at rt for 3 hr. It was quenched by slowly pouring the mixture into a vigorously stirred solution of sodium bicarbonate at 0 °C. After the bubbling stopped, the aqueous phase was extracted twice with $CH_2Cl_2$ (50 mL). The organic layer was washed with brine and concentrated to give an orange syrup. Silica column chromatography (eluent: 100% pentanes, $R_f$ = 0.27) yielded the product as a white solid (283 mg, 43%). The product's spectroscopic characterization was consistent with published data (*Li et al., 2016*; *Reibel et al., 2015*).

**¹H NMR** (400 MHz, CDCl₃) δ 5.39 (d, $J$ = 4.9, 1 H), 4.67–4.13 (dm, $^2J_{H-F}$ = 50.4 Hz, 1 H), 2.44 (t, $J$ = 7.0, 2 H), 2.10–1.92 (m, 3 H), 1.93–1.78 (m, 2 H), 1.77–1.63 (m, 1 H), 1.63–0.80 (m, 32 H), 0.69 (s, 3 H). **¹⁹F NMR** (377 MHz, CDCl₃) δ −167.82 (dm, $^2J_{F-H}$ = 50.4 Hz). **¹³C NMR** (101 MHz, CDCl₃) δ 139.50 (d, $J$ = 12.6 Hz), 123.16 (d, $J$ = 1.3 Hz), 92.98 (d, $J$ = 174.1 Hz), 56.88, 56.33, 50.17 (d, $J$ = 1.8 Hz), 42.49, 39.91, 39.69, 39.57 (d, $J$ = 19.3 Hz), 36.69 (d, $J$ = 1.2 Hz), 36.53 (d, $J$ = 10.8 Hz), 36.36, 35.95, 32.09 (d, $J$ = 1.1 Hz), 32.03, 28.95 (d, $J$ = 17.5 Hz), 28.39, 28.18, 24.45, 24.01, 22.98, 22.73, 21.29, 19.47, 18.89, 12.02. **HRMS (EI)**: Calcd. For C₂₇H₄₅F: 388.3505; Found: 388.3506.

## Computational rigidity-transmission allostery analysis

The fully active state of A₂ₐR in complex with Gₛαβγ, NECA and GDP was constructed, equilibrated and relaxed in a 1 µs MD simulation in 4:1 POPC:cholesterol extended membrane as previously described (*Huang et al., 2021*). This model was used to probe agonist-induced allosteric communication in the A₂ₐR-Gαβγ complex with rigidity-transmission allostery (RTA) algorithm, whose details have been previously described (*Huang et al., 2021*; *Ye et al., 2018*). The RTA algorithm is a computational method based on mathematical rigidity theory, which predicts how perturbations of conformational rigidity and flexibility (conformational degrees of freedom) at one site transmit across a protein or a protein complex to modify degrees of freedom at other distant sites (*Sljoka, 2021*). Here, RTA was applied to examine the allosteric pathways between the orthosteric pocket and distal regions in the A₂ₐR-Gαβγ complex with and without cholesterol. We quantified the available conformational degrees of freedom at every residue before and after rigidification of the agonist NECA. The change in degrees of freedom was then extracted for each residue, which represents the extent of allosteric transmission from the orthosteric pocket. In the presence of cholesterols, the analysis was carried out as previously described (*Huang et al., 2021*). To measure the impact of cholesterol on allosteric communication, the same analysis was repeated upon removal of all seven cholesterols found within 6 Å of the receptor.

## Acknowledgements

This work was supported by the CIHR Operating Grant MOP-43998 to RSP; SKH was supported by Alexander Graham Bell Canada Graduate Scholarship-Doctoral from NSERC; AS was supported by CREST, Japan Science and Technology Agency (JST), Japan JPMJCR1402; RJP was supported by QEII FE Beamish Graduate Scholarship in Science and Technology. Special thanks to Dmitry Pichugin for NMR spectrometer maintenance. We gratefully acknowledge discussions with colleagues (Dr. Lauren May and Dr. Jo-Anne Baltos at Monash University and Dr. Sebastian Furness at the University of Queensland) regarding cholesterol allostery in the cell, which helped shape this in vitro study.

## Additional information

### Funding

| Funder | Grant reference number | Author |
|---|---|---|
| Canadian Institutes of Health Research | MOP-43998 | R Scott Prosser |
| Japan Science and Technology Agency | JPMJCR1402 | Adnan Sljoka |
| Alexander Graham Bell Canada Graduate Scholarship-Doctoral | | Shuya Kate Huang |
| Alexander Graham Bell Canada Graduate Scholarship-Doctoral | | Reizel J Pejana |
| CREST | | Adnan Sljoka |

| Funder | Grant reference number | Author |
|---|---|---|
| QEII FE Beamish Graduate Scholarship in Science and Technology | | Reizel J Pejana |

The funders had no role in study design, data collection and interpretation, or the decision to submit the work for publication.

## Author contributions
Shuya Kate Huang, Conceptualization, Data curation, Formal analysis, Investigation, Methodology, Visualization, Writing – original draft, Writing – review and editing; Omar Almurad, Reizel J Pejana, Formal analysis, Investigation, Writing – review and editing; Zachary A Morrison, Investigation, Writing – review and editing; Aditya Pandey, Louis-Philippe Picard, Investigation; Mark Nitz, Supervision, Writing – review and editing; Adnan Sljoka, Formal analysis, Funding acquisition, Investigation, Software, Writing – review and editing; R Scott Prosser, Conceptualization, Funding acquisition, Supervision, Writing – original draft, Writing – review and editing

## Author ORCIDs
Shuya Kate Huang (iD) http://orcid.org/0000-0003-0637-4313
Adnan Sljoka (iD) http://orcid.org/0000-0002-2398-9523
R Scott Prosser (iD) http://orcid.org/0000-0001-9351-178X

## Decision letter and Author response
Decision letter https://doi.org/10.7554/eLife.73901.sa1
Author response https://doi.org/10.7554/eLife.73901.sa2

# Additional files

## Supplementary files
• Transparent reporting form

## Data availability
Source data files are provided for Figures 2, 5, 7b, 7c, Figure 2-figure supplement 1 (Figure S1 in the original bioRxiv preprint), and Figure 6-figure supplement 1 (Figure S3 in the original bioRxiv preprint). These are deposited in the public repository Dryad, available at https://doi.org/10.5061/dryad.9w0vt4bgw.

The following dataset was generated:

| Author(s) | Year | Dataset title | Dataset URL | Database and Identifier |
|---|---|---|---|---|
| Huang SK, Prosser RS | 2021 | Source data files for the manuscript "Allosteric modulation of the adenosine A2A receptor by cholesterol" | https://datadryad.org/stash/share/Oi7lwL3IPop2-O40r0THtwuUzFHdtC4CYu-ANfouygQ | Dryad Digital Repository, 10.5061/dryad.9w0vt4bgw |

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
