## [Editor Report]

Cholesterol has long been known to have significant effects on G protein-coupled receptor (GPCR) ligand binding properties and stability, and cholesterol/GPCR interactions have frequently been observed in high-resolution structures. However, relatively limited biophysical work has been done to investigate the mechanistic basis for cholesterol's effects. This manuscript describes the use of a sensitive 19F NMR probe to monitor conformational equilibria in a prototypical GPCR, the A2a adenosine receptor. These experiments, together with data from other NMR experiments, computational analysis, and G protein assays, show that the subtle effects of cholesterol derive in large part from modulation of membrane biophysical properties, in contrast to conventional allosteric modulators that exert their effects through direct long-lived receptor binding.

---

## [Decision Letter]

**Decision letter after peer review:**

Thank you for submitting your article "Allosteric modulation of the adenosine A_2A_ receptor by cholesterol" for consideration by *eLife*. Your article has been reviewed by 3 peer reviewers, and the evaluation has been overseen by a Reviewing Editor and Volker Dötsch as the Senior Editor. The following individual involved in review of your submission has agreed to reveal their identity: Paul Schanda (Reviewer #2).

Essential revisions:

Overall, the reviewers found the manuscript interesting and timely, but raised concerns that would need to be addressed prior to publication. Key points include:

1. The nanodisc system is an appealing membrane mimetic, but like any mimetic system it is an imperfect replacement for a true biological membrane. It would be helpful to more clearly delineate the limitations of this approach in the text.

2. The interpretation of pressure NMR experiments appears to rely on assumptions that may not be entirely justified, including: a) the assumption that the effects observed are independent of direct effects on the protein itself and, b) the assumption that previous data on effects of pressure on membrane thickness in liposomes applies equally to nanodiscs.

3. The ^19^F cholesterol experiments have several potential confounding factors (see detailed reviews below). These points should be discussed in the text or addressed with additional experiments.

*Reviewer #1 (Recommendations for the authors):*

I have no specific suggestions to the authors other than to consider the effect of F-chol concentration in the empty nanodiscs on their NMR spectrum (Figure 6) which impacts their discussion of Figure S4, perhaps in a non-trivial manner. For example, it appears that the Figure S4 spectra have been normalized to the same peak height, but that the noise in the NECA/MiniG spectrum is substantially higher than in the ZM241385 spectrum, which could reflect differences in F7-chol concentration (right bottom panel) which could be responsible for the small chemical shift differences.

*Reviewer #2 (Recommendations for the authors):*

Page 16 states that the pressure NMR avoid potential complications with specific lipid-receptor interactions. That's possibly true, but the high pressure, exerted onto a disc, introduces a set of other complications.

In Materials and methods, first section (expression), the reference to "Franz Hagn, Manuel Etzkorn, 2013) does not seem to be correct.

Methods: The NMR samples contained trifluoroacetate, a compound often used to perturb protein structure. This should be commented. Moreover, the sentence on page 26 (bottom line) says that the samples contained TFA or fluroacetate. Which one was in which sample? Why these two, why not always the same?

*Reviewer #3 (Recommendations for the authors):*

My only concerns are with the ^19^F NMR titration data of Figure 6, wherein the NMR spectra from fluorinated cholesterol analogs are shown as a function of both analog concentration and the presence or absence of the receptor. First, for the data shown, the concentration of the receptor (when present) should be given in the figure caption and in mol% units (same units as for the cholesterol analogs). This data would be much more convincing if full titrations of ligand (i.e., cholesterol analogs) by the receptor had been carried out. If possible, such titration data should be collected and presented and analyzed using standard models for ligand binding to a receptor. The authors seem to be making the claim that because ligand binding appears to be in the rapid exchange NMR regime that the binding is therefore non-specific. However, lots of proteins bind ligands specifically and yet with only modest affinity and on/off rates rapid enough to fulfill "fast exchange NMR regime" conditions. I also note that some of the data of panel B is odd and difficult to explain, such as the differences between the receptor-free 6 mol% and 9 mol% spectra. I wonder of the orientation of a part of the cholesterol analog could be flipped, placing the -OH group in the membrane interior and the fluorinated tail at the membrane-water interface?

---

## [Author Response]

Essential revisions:Overall, the reviewers found the manuscript interesting and timely, but raised concerns that would need to be addressed prior to publication. Key points include:1. The nanodisc system is an appealing membrane mimetic, but like any mimetic system it is an imperfect replacement for a true biological membrane. It would be helpful to more clearly delineate the limitations of this approach in the text.

We added a paragraph on nanodiscs and discussed the advantages and disadvantages of such a system. We now write: “In this study, we set out to investigate both the magnitude and origin of the allosteric interplay between cholesterol and A_2A_R in phospholipid bilayers, using an identical model lipid bilayer system for all functional assays and biophysical experiments. Nanodiscs have been used extensively in functional and structural characterization of complex membrane proteins (Sligar and Denisov, 2021). The protocols used in the current experiments generated monodisperse 8 nm diameter nanodiscs containing a single receptor and roughly 35-40 lipids per leaflet in addition to cholesterol (Hagn et al., 2013; Huang et al., 2021). This is a reductionist system featuring a single receptor surrounded by a fluid POPC/POPG lipid bilayer, with 1-5 cholesterol molecules per leaflet across the concentration range that was investigated. Thus, cholesterol-mediated lateral phase separation, or receptor oligomerization are excluded in this analysis. On the other hand, the receptor can be complexed with heterotrimeric G protein, and the role of cholesterol in modulating the receptor’s state distribution and G protein coupling can be studied with exquisite sensitivity.”

2. The interpretation of pressure NMR experiments appears to rely on assumptions that may not be entirely justified, including: a) the assumption that the effects observed are independent of direct effects on the protein itself and, b) the assumption that previous data on effects of pressure on membrane thickness in liposomes applies equally to nanodiscs.

In comparison to water and proteins, membranes are far more compressible and actually fascinatingly soft systems. Thus, pressures of 100-2000 bar would act predominantly to order lipids in bilayers, be they in liposomes, bicelles, or nanodiscs. We’ve included additional references to this vast field of literature. The ^19^F NMR studies with the pressures used in this study also show very clear shifts in known activation signatures and a stabilization of the A_1_ state. We have observed at higher temperatures (over periods of weeks to months) protein unfolding. In this case, we lose ligand sensitivity and the TM6 ^19^F NMR resonances shift upfield to a single resonance (unpublished). We also routinely use pressures in the range of 1300-2100 bar during receptor purification to achieve cell lysis, which has had no observable effects on the protein. We don’t want to detract from the body of the paper with a lengthy discussion here but we have added a comment in the Discussion section.

3. The ^19^F cholesterol experiments have several potential confounding factors (see detailed reviews below). These points should be discussed in the text or addressed with additional experiments.

The main point is that no binding isotherm, using two different labels, ever identified a bound signature and only gave relatively small shifts (indicative of fast exchange). Therefore, our conclusion that the effects are consistent with transient interactions and no long-lived cholesterol bound states holds. It is true that equivalent shifts were observed with for example inverse agonist and agonist in some cases and we agree with the opinion of reviewer 1 that this could reflect multiple binding sites. We have elaborated on this in the discussion.

Reviewer #1 (Recommendations for the authors):I have no specific suggestions to the authors other than to consider the effect of F-chol concentration in the empty nanodiscs on their NMR spectrum (Figure 6) which impacts their discussion of Figure S4, perhaps in a non-trivial manner. For example, it appears that the Figure S4 spectra have been normalized to the same peak height, but that the noise in the NECA/MiniG spectrum is substantially higher than in the ZM241385 spectrum, which could reflect differences in F7-chol concentration (right bottom panel) which could be responsible for the small chemical shift differences.

We thank the reviewer for his/her comments and suggestions. The lower signal/noise of the NECA/mini-G spectrum in Figure S4 relative to the other conditions is simply a result of a more dilute receptor sample. The concentration of F7-chol relative to lipids remains the same (2%). We’ve also altered the discussion slightly to account for the possibility that the small chemical shift perturbations with ligands and mini-G may be a result of weak cholesterol interactions at multiple binding sites.

Reviewer #2 (Recommendations for the authors):Page 16 states that the pressure NMR avoid potential complications with specific lipid-receptor interactions. That's possibly true, but the high pressure, exerted onto a disc, introduces a set of other complications.

The pressures used, which were comparable if not less than those used in for example (Lerch et al., PNAS 2020) were necessary to reach equilibrium shifts between active and inactive, while encompassing similar changes in hydrophobic thickness. As discussed above, the ^19^F NMR spectra are completely consistent with a fully functional receptor.

In Materials and methods, first section (expression), the reference to "Franz Hagn, Manuel Etzkorn, 2013) does not seem to be correct.

We thank the reviewer for spotting this and have fixed the reference.

Methods: The NMR samples contained trifluoroacetate, a compound often used to perturb protein structure. This should be commented. Moreover, the sentence on page 26 (bottom line) says that the samples contained TFA or fluroacetate. Which one was in which sample? Why these two, why not always the same?

We thank the reviewer for the suggestion and have now clarified this in the methods. TFA was used as a ^19^F chemical shift reference (-75.6 ppm) in small concentration (20 µM) in samples where we have done receptor NMR. It is ~15 ppm away from the BTFMA signal which is convenient as a reference and for setting the spectral width. We used fluoroacetate (-217 ppm) as a reference for ^19^F-chol NMR because the chemical shift of the CF_3_ groups in F7-chol (-76 to -78 ppm) is close to that of the TFA and given the broadness of the peaks we avoided using TFA in case of spectral overlap.

Reviewer #3 (Recommendations for the authors):My only concerns are with the ^19^F NMR titration data of Figure 6, wherein the NMR spectra from fluorinated cholesterol analogs are shown as a function of both analog concentration and the presence or absence of the receptor. First, for the data shown, the concentration of the receptor (when present) should be given in the figure caption and in mol% units (same units as for the cholesterol analogs). This data would be much more convincing if full titrations of ligand (i.e., cholesterol analogs) by the receptor had been carried out. If possible, such titration data should be collected and presented and analyzed using standard models for ligand binding to a receptor. The authors seem to be making the claim that because ligand binding appears to be in the rapid exchange NMR regime that the binding is therefore non-specific. However, lots of proteins bind ligands specifically and yet with only modest affinity and on/off rates rapid enough to fulfill "fast exchange NMR regime" conditions. I also note that some of the data of panel B is odd and difficult to explain, such as the differences between the receptor-free 6 mol% and 9 mol% spectra. I wonder of the orientation of a part of the cholesterol analog could be flipped, placing the -OH group in the membrane interior and the fluorinated tail at the membrane-water interface?

This is an issue that came up with all reviewers. We’ve tried to clarify that there is no evidence for a tightly bound cholesterol species in slow exchange. The reason this is somewhat relevant is that the cooperative activation events that take place across an allosteric network are associated with millisecond timescale motions. So if cholesterol was exchanging on a faster timescale, it is harder to imagine how it would play a prominent role. It is conceivable that the bulky (CF_3_)_2_CF group may adopt interesting orientations within the lipid bilayer. But given its hydrophobicity we think that it is unlikely for the F7-chol to be flipped.